# [Re] On the reproducibility of "Discovering and Mitigating Visual Biases through Keyword Explanation"

## Abstract

A computer vision model in machine learning is only as good as its training data (Moseley, 2024). Visual biases and spurious correlations in training data can manifest in model decision-making, potentially leading to discrimination against sensitive groups (Mehrabi et al., 2021). To address this issue, various methods have been proposed to automate bias discovery and use these biases to train bias-aware computer vision models. However, one major drawback of these methods is their lack of transparency, as the discovered biases are often not human-interpretable. To overcome this, Kim et al. introduced a Bias-to-Text (B2T) framework that identifies visual biases as keywords and expresses them in natural language. This paper aims to reproduce their findings and expand upon the evaluation methods. The central claims that the authors make are that B2T (i) can discover both novel and known biases, (ii) can facilitate debiased training of image classifiers and (iii) can be deployed across different classifier architectures, such as Transformer- and convolutional neural network (CNN)-based models. We successfully reproduce their main claims and extend their findings by analyzing whether novel bias keywords discovered by B2T represent actual biases. Additionally, we conduct further robustness experiments, leading us to conclude that the framework not only discovers biases in data, but also is sensitive to changes in the underlying classification model, highlighting a future research direction. Our code is publicly available at `https://anonymous.4open.science/r/B2T-Repr-898B`.

## 1 Introduction

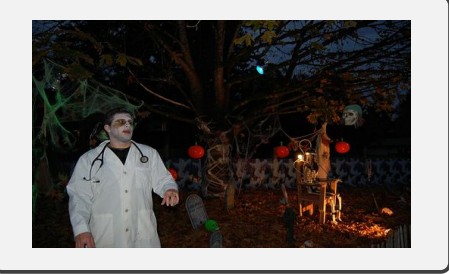

Figure 1: **Example of bias**. The Bias-to-Text (B2T) framework discovers visual biases in image data. As shown here, B2T detects the "haunted house" bias for the "stethoscope" class. This image has a high CLIP similarity score with the image, indicating that it is strongly associated with this specific bias.

Artificial intelligence is facing a reproducibility crisis (Hutson, 2018). In many cases, the results either heavily depend on specific experiment conditions, or the source code is simply not provided or complete. This emphasizes the importance of rigorous reproduction and validation of existing work. For this reproduction

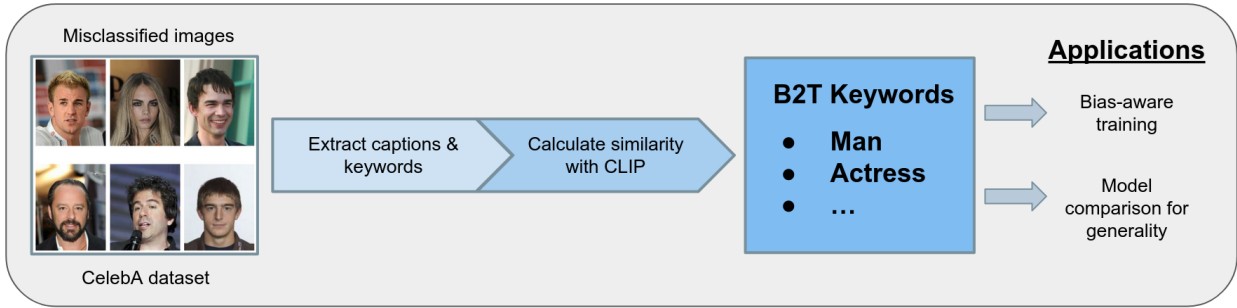

Figure 2: **Proposed framework**
.

study, we research the phenomenon of bias in image datasets. In practice, such datasets contain many forms of bias, raising fairness concerns when models trained on these datasets are deployed in the real world. For instance, a model naively trained to minimize average loss, i.e., an empirical risk minimization (ERM) model, tends to approximate a biased data distribution. As a result, it may produce biased outputs, potentially leading to discrimination against sensitive groups (Mehrabi et al., 2021).

To mitigate the adverse effects of bias in training data, various techniques for automated bias discovery have been proposed in the scientific literature (Bao & Barzilay, 2022; Zhang et al., 2022; Liu et al., 2021; Nam et al., 2020; Sohoni et al., 2020), enabling applications such as dataset debiasing or bias-aware learning. The effectiveness of a bias discovery method is typically evaluated by using the identified biases to train a model in a bias-aware manner, for example, using the group distributionally robust optimization (GDRO) loss function (Sagawa et al., 2019). The trained model is then tested on a dataset with ground-truth bias labels, where the performance is measured by two key metrics: worst-group accuracy, which reflects how well the model mitigates bias, and average accuracy, which indicates overall performance.

However, these state-of-the-art bias discovery methods usually suffer from one of the following limitations: a) they are not transparent, as identified visual biases are not human-interpretable, or b) they are supervised methods, requiring bias labels for training, which are rarely available (Kim et al., 2024). To address both limitations, Kim et al. (2024) recently introduced Bias-to-Text (B2T), a novel bias discovery framework that identifies visual bias as keywords expressed in natural language. B2T leverages a vision-language model to generate textual descriptions of images that a classifier failed to classify correctly, assuming that these images may be affected by bias. From these descriptions, the framework extracts keywords and assigns them a bias score (CLIP score) based on their similarity to misclassified images against correctly classified ones. A higher CLIP score suggests that a keyword represents a spurious correlation that makes certain images more prone to misclassification. It is important to note that B2T defines bias as the opposite from its conventional meaning. Typically, bias refers to data properties that favor a particular class. In contrast, B2T defines it as data properties that make a particular class harder to classify correctly.

Using B2T, Kim et al. identified both known biases and novel biases, such as "flower" in the "ant" class and "interior" in the "monastery" class within the ImageNet dataset (Deng et al., 2009). Additionally, they report that B2T, when combined with existing bias-aware learning techniques, achieves state-of-the-art performance on both worst-group and average accuracy metrics.

The aim of this paper is to reproduce and extend their findings in the following ways:

- **Reproduce the main results of the original paper**. We assess the validity of the findings, adhering to the provided code whenever possible. However, the methodology underpinning most results is insufficiently documented, making it difficult to evaluate their significance. In this work, we address these gaps by offering clear procedures where they were not detailed before.

- **Extend the results**. We extend the original results in two key ways:

- *Enhanced novel bias discovery.* We not only detect novel biases in the *ImageNet* Deng et al. (2009) dataset but also conduct additional experiments to verify that the bias keywords represent actual biases. This provides a more rigorous evaluation of B2T's effectiveness in identifying novel biases.
  - *Robustness evaluation.* We evaluate the robustness of B2T across two dimensions. Firstly, we examine how B2T responds to variations in input data sampled from the same distribution, evaluating its stability under fluctuations in the input data. Secondly, we test B2T's performance when the underlying classification model is swapped, ensuring that its effectiveness is not model-dependent.

- **Extend, fix, and clean code**. We provide a refactored codebase, forked from the original repository, which includes all code necessary to reproduce the results in this work. The refactored codebase features debugged code, improved organization, and clearer documentation.

In the following sections, we outline our scope of reproducibility in section 2, and describe the methodology in section 3. Thereafter, we present our results in section 4. Lastly, we conclude with a discussion of our findings, comparing them to the original paper, and examining their broader implications in section 5.

## 2 Scope of reproducibility

Fairness in artificial intelligence (AI) systems has recently gained significant attention, as these systems are often deployed in sensitive environments (Mehrabi et al., 2021). In many cases, fairness concerns arise from data imbalance or representation bias within the utilized datasets, potentially inducing model failures (Kim et al., 2024). In computer vision this is often an issue of visual bias, which are not human-interpretable. To mitigate these issues, various visual bias discovery methods have been proposed, enabling the identification and elimination of biases either directly within the data or at the model level, thereby enhancing the fairness and reliability of AI systems. However, as previously mentioned, these methods typically either lack transparency or have limited applicability in real-world scenarios as they require supervised training data. B2T addresses both points by providing bias keywords in natural language and not needing ground-truth bias labels. By making the bias discovery process more transparent, the subsequent bias elimination procedure becomes more transparent by extension.

After visual bias discovery, B2T-generated bias keywords are used to infer a bias label for each image in the dataset using CLIP. These predicted bias labels can then be seamlessly integrated into the distributionally robust optimization (DRO), also referred to as group DRO (GDRO), a framework to train a debiased model.

Overall, the original paper stated the following central claims:

1. **Claim 1 (bias discovery):** B2T can successfully discover both known and novel biases, including gender bias, background bias, contextual bias, and distribution shifts, while outperforming previous state-of-the-art bias discovery methods. Additionally, B2T explains these biases using keywords expressed in natural language.

2. **Claim 2 (bias elimination):** The bias keywords obtained from B2T Keywords can be used to train a debiased model with the DRO method, referred to as DRO-B2T. This approach outperforms state-of-the-art debiasing methods in terms of both worst-group and average accuracy. Moreover, DRO-B2T surpasses standard DRO with ground-truth bias labels across both accuracy metrics.

3. **Claim 3 (generality):** B2T can be deployed with various classifier models, such as Transformer- or CNN-based architectures, while still retrieving meaningful bias keywords.

In addition to the claims stated above, the authors propose and discuss potential applications for B2T, such as improving CLIP zero-shot prompting and label diagnosis, among other things. However, these applications are numerous, discussed only briefly, and are not central to evaluating the effectiveness proposed B2T framework. Therefore, we focus on presenting a study that validates the core contribution of the original paper: the B2T framework.

# 3 Methodology

In this section, we present an overview of the methodology. Whenever possible, we adhere to the documentation and code provided by the authors[1]. In cases where the description of the method or code is incomplete or unavailable, we detail our design and implementation decisions.

## 3.1 Datasets

The original work considers two types of datasets: those for which bias labels are known and those for which they are not. The former allows for the verification of biases discovered by B2T on standard datasets commonly used in the field of bias discovery. On the other hand, the latter enables the discovery of novel biases within standard datasets used more broadly in the field of AI.

- The first dataset that includes bias labels is the *Waterbirds* dataset (Sagawa et al., 2019), containing $11,708$ images of birds. Each image is labeled either as a waterbird (e.g., a duck) or a landbird (e.g., a hawk). Background biases are artificially introduced by editing some waterbird samples to appear against land backgrounds and some landbirds samples against water backgrounds. These modifications make these samples more challenging for the model to classify correctly, thereby introducing background bias.

- The *CelebFaces Attributes Dataset* (referred to as *CelebA*) contains over $200,000$ images (Liu et al., 2015) of celebrities. It is known that the blond-haired class exhibits a gender bias towards women. According to the B2T definition of bias, the correct bias keyword for the blond class would be "man".

- The *ImageNet-R* Hendrycks et al. (2021) and *ImageNet-C* Hendrycks & Dietterich (2019) datasets represent distribution shifts, which are considered biases. The authors use subsets of $30,000$ and $50,000$ images, respectively. For *ImageNet-C*, which includes varying levels of corruption, $25,000$ images were uniformly sampled from each severity level, with an additional $25,000$ from the uncorrupted *ImageNet* dataset.

* The dataset without bias labels is *ImageNet* (Deng et al., 2009), which consists of $1,281,167$ images distributed over $1,000$ classes.

For each dataset, we apply the same preprocessing steps as described in the original paper. Although the original work also considered the *Dollar Street* dataset Rojas et al. (2022), we focus solely on the aforementioned datasets due to the limited research conducted on the *Dollar Street* dataset by the original authors.

## 3.2 Model descriptions

To discover biases, the B2T framework employs the following models at each step of the process:

- **Captioning**: Textual descriptions of images are generated using the pretrained *ClipCap* captioning model provided by Mokady et al.. This model uses a *CLIP* (Radford et al., 2021) embedding of an image as a prefix to autoregressive text completion with *GPT-2* (Radford et al., 2019).

- **Image classification**: B2T extracts keywords from the captions of incorrectly classified images, where classification is performed by a given image classifier (Vapnik, 1991). Following the B2T paper, the classifiers of choice for the *Waterbirds*, *CelebA*, and *ImageNet* datasets are pretrained *ResNet-50* models (He et al., 2016; maintainers & contributors, 2016). Additionally, we evaluate a pretrained *ViT-B/16* model (Alexey, 2020; maintainers & contributors, 2016) on the *ImageNet* dataset. Notably, the authors of the original paper also present a ViT versus ResNet comparison on ImageNet. However, their comparison aims to illustrate that ViT-based models are more successful at predicting abstract keywords like "work out" than CNN-based models. In contrast, we make a complete comparison between discovered keywords when using either type of model.

---

[1]Code: `https://github.com/alinlab/b2t`

- **Keyword extraction**: After classification, potential bias keywords are obtained from the concatenated captions of the misclassified images using the *YAKE* algorithm (Campos et al., 2020), which is a state-of-the-art non-parametric and unsupervised keyword extraction technique.

- **Keyword scoring**: To score a potential bias keyword, the average CLIP similarity score (i.e., inner product between an image and keyword embedding, scaled to be between 0 and 100) of the correctly classified images is subtracted from the average CLIP score of the misclassified images. The authors of the original paper refer to this quantity as "CLIP score", which should not be confused with the CLIP similarity score. Potential bias keywords with a positive CLIP score are marked as bias keywords, where a higher value indicates a higher level of bias.

### 3.3 Hyperparameters

For our reproduction, we reuse the hyperparameter values given in the codebase to ensure consistency and reproducibility of the results. Each experiment is repeated three times using a different random seed for each run.

### 3.4 Experimental setup and code

In this subsection, we provide the details necessary to replicate our experimental setup. For implementation details, please refer to our repository.

#### 3.4.1 Claim 1 (bias discovery)

To verify the effectiveness of B2T at discovering biases, we conduct a series of experiments focusing on key aspects of bias discovery. Firstly, we *visualize* examples of keyword discovery for a small selection of classes to illustrate what biases B2T detects. Secondly, we *verify* the quality of the extracted keywords using statistical analysis.

**Example visualization experiment:** To illustrate example bias keywords detected by B2T, the original authors display misclassified images alongside corresponding bias keywords and captions. However, they present no procedure for selecting these examples or matching specific sample images with the corresponding bias keywords. To replicate a similar result, we extract keywords for the same selection of classes as the original paper. For each class, we display the keyword with the highest CLIP score, the image within that class that has the highest CLIP similarity score with the keyword, and the image's caption. This approach ensures that the resulting figure highlights the most dominant examples of detected bias for the selected classes.

**Bias verification experiment:** To verify whether a discovered bias keyword represents an actual bias, we reproduce the CLIP score analyses from the original paper on the *Waterbirds* and *CelebA* dataset. The original authors assess the validity of a bias keyword by comparing its CLIP score with the receiver operating characteristic (ROC) curve for the bias subgroups that correspond with that keyword, using the area under the curve (AUC) (Fawcett, 2006) metric to quantify performance. If a bias keyword represents an actual bias, the subgroup of images affected by that bias is expected to perform worse relative to unbiased subgroups.

However, the original authors are unclear about the procedure used to assign images to keyword subgroups. They mention that images with a high CLIP similarity score to a keyword are assigned to the corresponding subgroup. However, this description leaves room for multiple possible assignment strategies, such as percentile-based (e.g., selecting the top n%) or mean thresholding. To ensure that only images with a comparatively high CLIP score are assigned to a subgroup, we adopt mean thresholding for subgroup assignment.

As an extension, we additionally evaluate B2T's effectiveness in discovering novel biases by verifying the biases detected within the *ImageNet* dataset. However, the aforementioned analysis does not directly apply to the multi-class *ImageNet* dataset, since ROC analysis assumes a binary setting. Therefore, we transform the *ImageNet* classification problem into a binary one by adopting a one-versus-the-rest scheme, where the correct class is treated as the "true" label, and all other classes as "false".

### 3.4.2 Claim 2 (bias elimination)

**DRO training experiment:** To evaluate B2T's effectiveness in training a debiased model, we reproduce the DRO-B2T training process - i.e., DRO training of a ResNet-50 model using bias labels inferred from B2T-generated keywords - on the *Waterbirds* and *CelebA* datasets, following the code provided by the original authors. For a more detailed description of inferring bias labels, refer to Appendix B. Additionally, we reproduce the DRO training using ground-truth bias labels. Finally, we assess the performance of both models using worst-group and average accuracy metrics.

### 3.4.3 Claim 3 (generality)

**Robustness evaluation experiment:** As another extension, we examine the robustness of the B2T framework by testing the consistency of the discovered keywords with regard to:

a) *Variations in the underlying classification model.* The ultimate goal of B2T is to discover biases in *data* using a pipeline consisting of four sequential steps: captioning, classification, keyword extraction, and keyword scoring. For the bias discovery process to remain unbiased itself, a key underlying assumption is that each step in the procedure does not introduce additional biases into the resulting bias keywords.

Since, in the B2T sense, bias is defined as a data property that makes data points more difficult to classify correctly, the concept of bias is inherently linked to the classification function used. Therefore, as an additional experiment, we evaluate B2T's robustness to variations in the classification model by comparing the resulting bias keywords when using a ResNet-50 and ViT-B/16 model on the *ImageNet* dataset.

b) *Input data sampled from the same distribution.* While the original paper considers datasets where hundreds to thousands of images are available per class, a potential area of interest lies in small datasets with fewer than one hundred images per class. To explore B2T's effectiveness with such small datasets, we simulate bias discovery on different partitions sampled from the same distribution and evaluate the consistency of the resulting bias keywords. If B2T is robust with small datasets, we expect the detected keywords to remain consistent despite fluctuations in input data.

Specifically, we test the robustness of B2T on the two novel biases that the original authors claim to have discovered within the *ImageNet* dataset: "flower" bias in the "ant" class and "interior" bias in the "monastery" class. To assess input data robustness, we conduct experiments on 10 partitions of a 500 image sample.

### 3.5 Computational requirements

We use the Dutch national supercomputer Snellius[2] to conduct our experiments, utilizing NVIDIA A100-SXM4-40GB and NVIDIA H100 GPUs. To measure the total carbon equivalent, runtime, and memory usage of each experiment, we employ Energy Aware Runtime[3] (EAR). We report measurements only for DRO training, excluding evaluation and B2T bias discovery, as training is significantly more energy-intensive.

## 4 Results

### 4.1 Claim 1 (bias discovery)

The first main claim of the original paper is that B2T can successfully detect known and novel biases.

**Example visualization experiment:** Table 1 supports the claim that B2T can successfully detect known biases. For instance, B2T accurately identifies gender bias in the *CelebA* dataset, background bias in the *Waterbirds* dataset, and the correct distribution shifts for the *ImageNet-R* dataset. In the case of the

---

[2]https://hpc.uva.nl/Why-/article/115/Dutch-National-Supercomputer-Snellius
[3]https://gitlab.bsc.es/ear_team/ear/-/wikis/home

*ImageNet-C* dataset for the snow/frost corruptions, B2T incorrectly predicts "rain" as a bias. However, this prediction is semantically very similar to the correct corruption bias.

Interestingly, B2T also discovers the keyword "seagull flies" for the landbird class in the *Waterbirds* dataset. Although this keyword is not part of the ground-truth bias labels, the corresponding image does contain a flying seagull. This suggests that B2T may have detected a labeling error rather than a background bias.

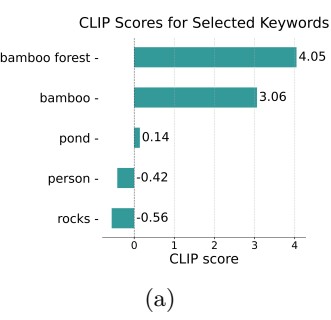
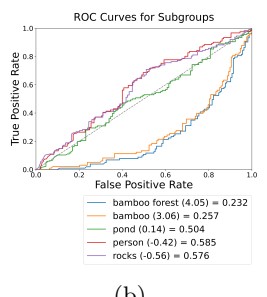
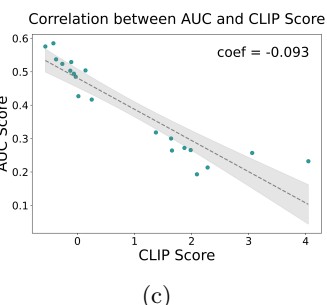

(a)            (b)            (c)

Figure 3: **CLIP score validation** (waterbird class). (a): Bar chart ranking keywords by CLIP similarity. (b): ROC curves for subgroups defined by keywords, with each keyword's CLIP score and AUC in the legend. AUC $\approx 0.5$ indicates no bias, with lower values indicating a stronger bias. (c): Scatter plot of CLIP scores versus AUC, illustrating that higher scores are associated with higher biases.

**Bias verification experiment:** Firstly, our results for the bias verification experiment support the claim that B2T can detect known biases. For instance, from Figure Figure 3, we can draw two critical observations: a) we further validate B2T's capability to detect known biases, as demonstrated by the "bamboo" and "bamboo forest" keywords, which correctly identify background bias; and b) we confirm that the detected bias keywords represent actual biases for the waterbird class, as the keywords with higher CLIP scores correlate with lower AUC scores. Moreover, as shown in Appendix A.2, these observations hold true for the other classes in the *Waterbirds* and *CelebA* datasets as well. Overall, our figures closely resemble those from the original paper.

Secondly, our results further support the claim that B2T can detect novel biases across various *ImageNet* classes. For instance, Figure 5 illustrates that B2T detects the novel "flower" bias in the "ant" class, replicating the finding reported in the original paper. The figure also demonstrates a correlation between higher CLIP scores and lower AUC scores, confirming that the bias keywords represent actual biases. Additionally, the evaluations of other *ImageNet* classes in Appendix A.2 reflect similar patterns, suggesting that B2T can consistently discover novel biases. However, we note that the correlations between CLIP and AUC scores for *ImageNet* classes are weaker compared to those observed in the *Waterbirds* dataset.

## 4.2 Claim 2 (bias elimination)

**DRO training experiment:** After bias discovery, B2T-generated keywords are used to infer bias labels for each image in the *Waterbirds* and *CelebA* datasets. Leveraging these labels for bias-aware training, we find that the DRO-B2T outperforms all other state-of-the-art methods considered, achieving superior results on both the worst-group and average accuracy metrics, as shown in Table 2. Notably, it even surpasses DRO training with ground-truth bias labels, suggesting that B2T-generated bias keywords may be more effective for debiasing than the human-made annotations for the *Waterbirds* and *CelebA* dataset. Although our accuracy scores differ slightly from those reported in the original work, our findings remain consistent with the original paper.

After rerunning bias discovery with our DRO-B2T model as the underlying classifier, we observe that fewer bias keywords are detected compared to the original ERM model, as shown in Table 3. Furthermore, the bias keywords that are discovered using DRO-B2T exhibit lower CLIP scores. These findings reinforce that the DRO-B2T model is indeed less biased than its ERM counterpart, which is consistent with the conclusions of the original paper.

Table 1: **Visualization of discovered biases.** Examples of biased keywords are displayed alongside their corresponding highest-scoring images based on CLIP similarity score, along with associated captions, actual labels, and predicted labels.

| | CelebA blond | | Waterbirds | | ImageNet-R | |
|---|---|---|---|---|---|---|
| **Keyword** | man | actress | bamboo forest | seagull flies | cartoon | tattoo |
| **Samples** | | | | | | |
| **Actual** | blond | not blond | waterbird | landbird | backpack | great white shark |
| **Pred.** | not blond | blond | landbird | waterbird | envelope | bathing cap |
| **Clip s. score** | 27.12 | 29.64 | 32.75 | 31.14 | 27.52 | 29.94 |
| **Caption** | person, a professor of biology, is a professor of biology. | actor – i love her hair and makeup. | biological species in a bamboo forest. | biological species in flight over the sea. | drawing of a face with a face mask. | tattoo on the back of the right arm. |

| | ImageNet-C snow/frost | | ImageNet | | | |
|---|---|---|---|---|---|---|
| **Keyword** | rain | rain | yellow flower | beam | haunted house | altar |
| **Samples** | | | | | | |
| **Actual** | airliner | American egret | ant | horizontal bar | stethoscope | monastery |
| **Pred.** | ski | dam | daisy | parallel bars | lab coat | altar |
| **Clip s. score** | 27.52 | 28.39 | 33.66 | 27.17 | 27.42 | 31.73 |
| **Caption** | airplane flying in the rain. | a water fountain in the forest. | a bee on a yellow flower. | the best way to measure the height of a pole is with a ruler. | person in the haunted house. | the altar of the church. |

Table 2: An overview of the worst-group and average accuracies for our debiased classifier (DRO-B2T), the debiased classifier (DRO-B2T) of Kim et al. (2024), and prior debiasing works.

| | | CelebA blond | | Waterbirds | |
|---|---|---|---|---|---|
| Method | GT | Worst | Avg. | Worst | Avg. |
| ERM | - | 47.7±2.1 | 94.9 | 62.6±0.3 | 97.3 |
| LfF | - | 77.2 | 85.1 | 78.0 | 91.2 |
| GEORGE | - | 54.9±1.9 | 94.6 | 76.2±2.0 | 95.7 |
| JTT | - | 81.5±1.7 | 88.1 | 83.8±1.2 | 89.3 |
| CNC | - | 88.8±0.9 | 89.9 | 88.5±0.3 | 90.9 |
| DRO-B2T (ours) | - | 89.1±0.3 | 92.0±1.2 | 88.7±1.6 | 93.1±0.1 |
| DRO-B2T (theirs) | - | **90.4±0.9** | 93.2 | **90.7±0.3** | 92.1 |
| DRO | ✓ | 88.47±0.33 | 91.9±0.6 | 88.3±1.2 | 93.4±0.1 |

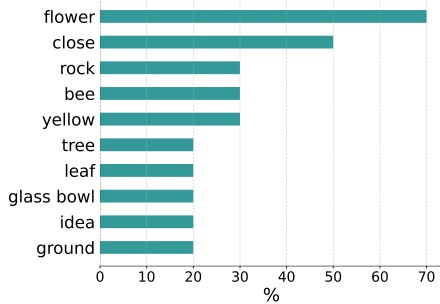

Relative frequencies of bias keywords

Figure 4: The data robustness evaluation for the *ImageNet* "ant" class. We split the class data into 10 partitions, and for each partition, we run B2T. The relative frequencies showcase the proportion of partitions for which a certain bias keyword was discovered.

Table 3: Comparison of the bias keywords discovered by ResNet-50 (i.e., ERM) and the keywords discovered by our trained DRO-B2T for the waterbird class. As expected, the DRO model is susceptible to significantly fewer biases.

| Bias keyword | ERM | DRO-B2T | Match |
|---|---|---|---|
| bamboo forest | 4.05 | - | - |
| bamboo | 3.06 | - | - |
| forest | 2.28 | 1.34 | ✓ |
| woods | 2.09 | - | - |
| rainforest | 1.98 | - | - |
| trees | 1.88 | - | - |
| garden | 1.66 | 1.11 | ✓ |
| tree | 1.64 | 1.23 | ✓ |
| bird of paradise | 1.38 | - | - |
| bird | - | 0.92 | - |
| wall | - | 0.48 | - |
| pond | 0.14 | 0.31 | ✓ |
| wild | 0.25 | 0.08 | ✓ |
| prey | 0.02 | 0.14 | ✓ |
| paradise | - | 0.02 | - |

Table 4: Keywords of the ImageNet monastery class according to the ViT-B/16 and ResNet-50 model.

| Bias keyword | ViT-B/16 | ResNet-50 | Match |
|---|---|---|---|
| ceiling | - | 1.80 | - |
| cathedral of person | - | 0.83 | - |
| inside the church | - | 0.69 | - |
| interior | 0.45 | 0.64 | ✓ |
| altar | 0.55 | 0.61 | ✓ |
| deity | - | 0.41 | - |
| bell | 0.25 | - | - |
| person | 0.19 | - | - |
| church of person | - | 0.17 | - |
| temple of person | 0.16 | - | - |
| library | - | 0.09 | - |
| view | 0.03 | - | - |

### 4.3 Claim 3. (generality)

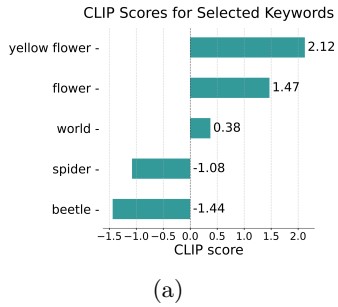
(a)

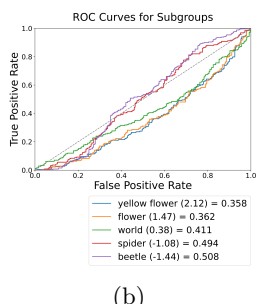
(b)

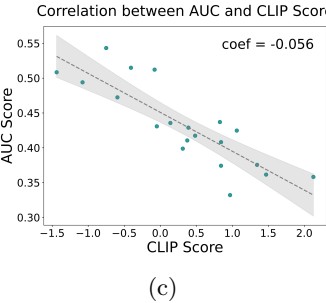
(c)

Figure 5: **CLIP score validation** (ant class). (a): Bar chart ranking keywords by CLIP similarity. (b): ROC curves for subgroups defined by keywords, with each keyword's CLIP score and AUC in the legend. AUC ≈ 0.5 indicates no bias, with lower values indicating a stronger bias. (c): Scatter plot of CLIP scores versus AUC, illustrating that higher scores are associated with higher biases.

**Robustness evaluation experiment:** We test B2T's robustness in two aspects:

a) *Variations in the underlying classification model.* We find that B2T-generated bias keywords are *not* consistent across changes in the underlying classification model. For example, the ViT and ResNet models discover a total of 23 unique bias keywords for the "ant" class, of which they share only 6 (i.e., 26%). Similarly, the models share 2 out of eleven 11 (i.e., 17%) of the keywords for the "monastery" class. Nonetheless, we still replicate the original finding, where B2T detects the "flower" keyword for the "ant" class and the "interior" keyword for the "monastery" class for both model types. However, due to the high disagreement between models, this could be attributed to coincidence.

b) *Input data sampled from the same distribution.* To evaluate B2T's effectiveness for small datasets, we assess its consistency across 10 partitions of 50 images, sampled from the same class distributions, as shown in Figure 4. We find that B2T-generated bias keywords are relatively consistent across different data partitions. For instance, despite the small size of the partitions, the original finding of the novel "flower" bias in the "ant" class is detected in 70% of the partitions. Apart from the dominant biases, many other keywords are discovered in only a small fraction of partitions, indicating they are less prevalent in the majority of partitions.

### 4.4 Emissions

Table 5 presents the memory and CO2 equivalent metrics for running the DRO experiments. As expected, DRO and DRO-B2T show similar carbon equivalence, memory usage, and runtime, since the training procedure remains the same; only the bias labels used differ.

Table 5: Power, runtime and memory usage of models across datasets for DRO training.

| Model | CelebA | | Waterbirds | | Runtime (hours) | |
|---|---|---|---|---|---|---|
| | $CO_2$eq (g) | Memory (GB) | $CO_2$eq (g) | Memory (GB) | CelebA | Waterbirds |
| DRO | 399 | 1.75 | 178 | 1.28 | 2.54 | 1.16 |
| DRO-B2T | 461 | 1.66 | 153 | 1.43 | 3.29 | 1.38 |

## 5 Discussion

### 5.1 Verifying the main claims

The objective of our study was to validate the recently proposed B2T method by Kim et al. for automated bias discovery in image data. To achieve this, we reproduced their results to assess the three central claims they make: a) B2T can effectively detect known and novel biases; b) B2T-generated bias keywords can be integrated with DRO to train a bias-aware model; and c) B2T is compatible with different underlying classifier architectures.

### 5.2 Claim 1 (bias discovery)

To verify their first claim - i.e., B2T can successfully discover known and novel biases - we retrieved bias keywords for various example classes using B2T. We then evaluated whether these keywords represented actual biases by inspecting the ROC curves of different biased subsets of the datasets. Our findings align with the original paper: B2T successfully identifies background bias in the *Waterbirds* dataset, gender bias in the *CelebA* dataset, and distribution shifts in the *ImageNet-C* and *ImageNet-R* datasets.

Additionally, we discover the same novel biases reported in the original work, like the "flower" bias in the "ant" class. As an extension, we confirmed that this and other novel biases identified within the *ImageNet* dataset correspond to actual biases, further supporting the first main claim. However, we observe that the correlation between CLIP and AUC scores for novel *ImageNet* biases is pronounced compared to, for example, biases in the *Waterbirds* dataset. This discrepancy may be attributed to the more overt nature of biases in *Waterbirds*, whereas biases in *ImageNet* are likely more subtle. Namely, biases for the *Waterbirds* generally receive higher CLIP scores than those in *ImageNet*, suggesting that the biases in *Waterbirds* are inherently stronger.

Within the domain of fairness in AI, these results position B2T as a competitive method for bias discovery. Not only does it successfully discover biases, but it also provides human-interpretable keywords and does not require supervised labels for training. This combination of transparency and practicality makes B2T a promising tool for real-world applications.

### 5.3 Claim 2 (bias elimination)

After obtaining bias keywords using B2T, we used these to infer bias labels for each image in the dataset as described by Kim et al.. These could then seamlessly be integrated into the DRO framework to train a debiased model, which we refer to as DRO-B2T. Our findings agree with the original work and demonstrate that DRO-B2T outperforms previous state-of-the-art debiasing methods on worst-group and average accuracy metrics, even outperforming DRO using ground-truth labels. This result suggests that B2T keywords are highly suitable for debiasing applications, potentially providing more relevant bias representations than human-annotated labels.

Moreover, DRO-B2T reduces both the number of bias keywords and their corresponding CLIP scores. This reduction implies that incorporating B2T-generated keywords into the DRO framework results in a more debiased classifier. However, it is important to note that this method has only been applied to the *Waterbirds* and *CelebA* datasets. Therefore, further research is necessary to assess the generalizability of the DRO-B2T method to other datasets.

Overall, we successfully reproduced the original findings on the effectiveness of DRO-B2T and validated it as an extension to the widely popular DRO framework, enabling debiasing in settings where bias labels are not available.

### 5.4 Claim 3 (generality)

*Variations in the underlying classification model.*
While the goal of B2T is to detect biases within data, it may also capture biases introduced by the underlying classification model. To assess the extent to which the identified bias keywords originate from data-related biases rather than model-induced biases, we compare B2T-generated bias keywords for a ViT-B/16 and a ResNet-50 model on the *ImageNet* "ant" and "monastery" class in Table 4 and Table 6 in Appendix C.

Our results reveal a severe discrepancy between the keywords when using either of the two models, with the models sharing fewer than a quarter of the keywords. We attribute this disagreement to the fact that the ViT- and CNN-based model may be susceptible to different types of bias, which, in turn, influence B2T's keyword extraction process. From these findings, we conclude that further research in necessary to determine the desirable properties of the underlying classifier for effective B2T performance.

*Input data sampled from the same distribution.*
Additionally, we explored the potential application of B2T on small datasets where fewer than 100 images are available per class. Specifically, we evaluated the robustness of B2T-extracted keywords by analyzing keywords for 10 partitions of 50 images each, sampled from the *ImageNet* "ant" class. Our results show that B2T successfully detects the novel bias keyword "flower" in the majority of partitions, indicating that B2T is robust for small datasets in this setting. However, further research is needed to fully understand B2T's behavior on small datasets, particularly regarding its sensitivity to outliers.

### 5.5 Communication with original authors

We contacted the authors via email to seek clarification on the undocumented or missing parts in the methodology, but we did not receive a timely response.

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

## A    Bias Verification

### A.1    Methodology

In validating the CLIP score, keywords for which the ROC curves are visualized are selected by ranking all keywords based on their CLIP score. From this ranking, the two highest-scoring, and two lowest-scoring keywords are chosen, along with one keyword near the middle of the distribution. This approach helps to minimize the biased selection of keywords introduced by manual selection and enhances the robustness of the evaluation.

### A.2    Additional Validations

To further assess how well the CLIP score validation generalizes across different classes with known or novel biases, we extend our analysis beyond the two classes presented in the main paper. Specifically, We present validations for the landbird class in the Waterbirds dataset (Figure 6), both classes in the CelebA dataset (Figure 7 and Figure 8), and two additional ImageNet classes, including stethoscope (Figure 9) and whale (Figure 10). By conducting these additional analyses we provide more evidence supporting the correlation between high CLIP similarity and biases, reinforcing the claim that B2T can effectively discover known and novel biases.

## B    Inferring bias labels with the CLIP zero-shot classifier

To infer bias or group labels, we provided the B2T keywords in the prompts of the CLIP zero-shot classifier. For the specific prompts, we used the prompt template provided by Kim et al. (2024), which assigns each bias keyword to each of the possible prompts. This process was done for both the *CelebA* and the *Waterbirds* dataset. For the *CelebA* dataset, we used "blond man" as the false positive rate, whereas for the *Waterbirds* dataset, we used "landbird on water background" and "waterbird on a land background" as the false positive rate for the "landbird" and "waterbird" classes respectively.

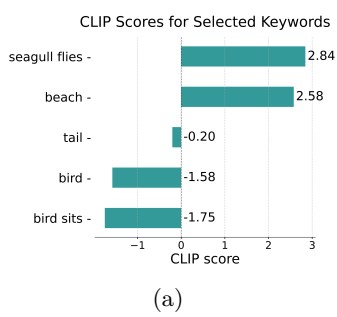
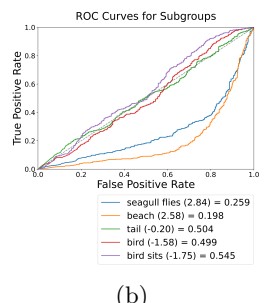
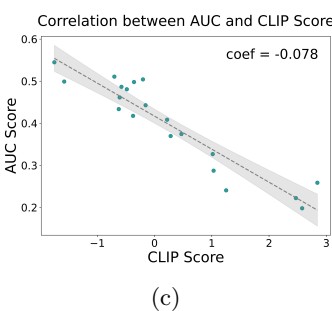

(a)            (b)            (c)

Figure 6: **CLIP score validation** (landbird class). We observe similar trends to the ones shown in the waterbird class. This shows that the validation generalizes to both classes in the Waterbirds dataset.

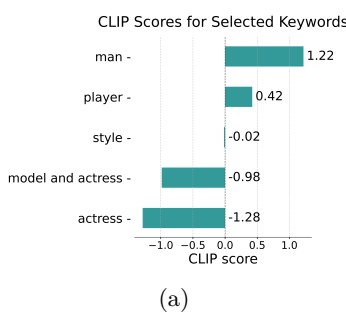
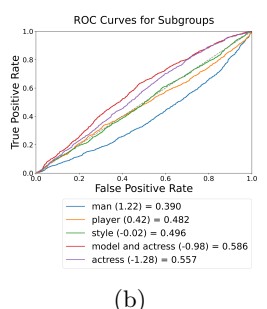
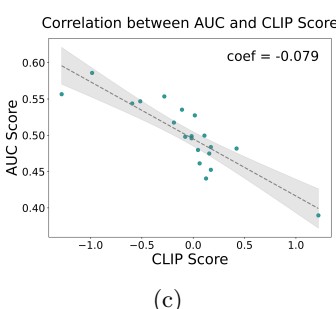

(a)            (b)            (c)

Figure 7: **CLIP score validation** (blond class). We observe similar trends to the ones shown in the waterbird class. This shows that the validation generalizes to the blond class in the CelebA dataset.

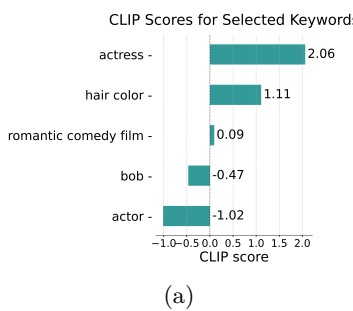
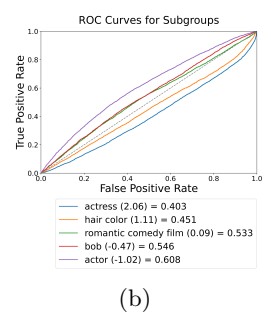
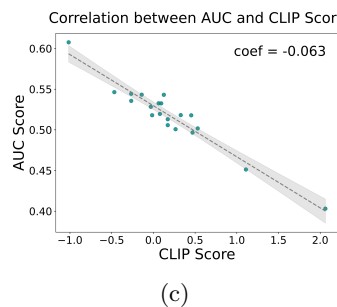

(a)            (b)            (c)

Figure 8: **CLIP score validation** (not blond class). We observe similar trends to the ones shown in the waterbird class. This shows that the validation generalizes to both classes in the CelebA dataset.

### B.1 ROC curves of the *CelebA* and Waterbirds datasets

In Figure 11, the ROC curves of the datasets *CelebA* and *Waterbirds* are plotted to visualize the impact of inferring bias labels after providing B2T keywords in CLIP zero-shot prompts. Overall, most ROC curves are reproduced from the paper of Kim et al. (2024), with the exception of class "waterbird". For "waterbird", the curve is more steep, which may be related to our difference in bias keywords and CLIP scores compared to theirs. For the *CelebA* "blond" and "landbird" classes, however, the curves closely align with the authors' results, which indicates that using bias keywords for the CLIP zero-shot classifier can infer or predict bias (group) labels. This explains the rapid incline of the true positive rate.

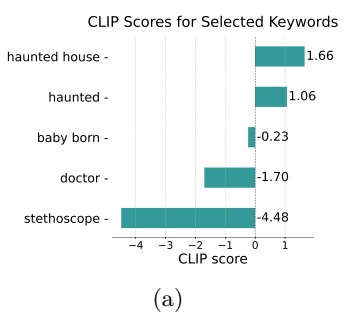

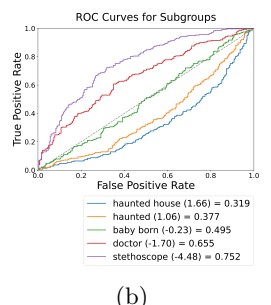

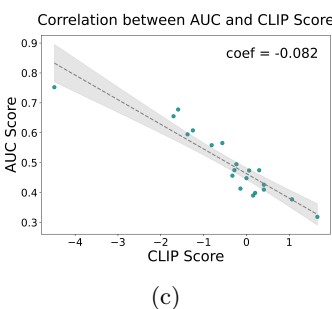

(a)                                        (b)                                        (c)

Figure 9: **CLIP score validation** (stethoscope class). We observe similar trends to the ones shown in the waterbird class. However, the correlation between CLIP score and AUC score is less evident. This can be attributed to potential misalignments in the methodology and the usage of a multiclass classification model. This analysis shows that the validation generalizes to the stethoscope class, but it raises the question of whether this is also true for the entire *ImageNet* dataset.

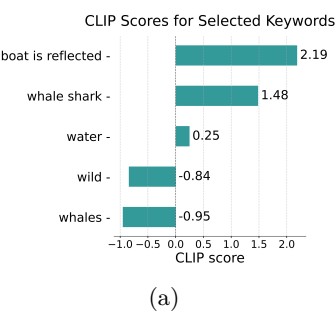

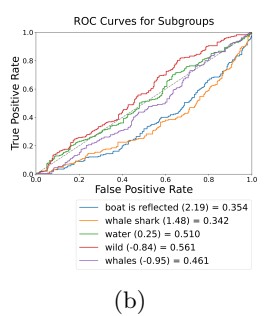

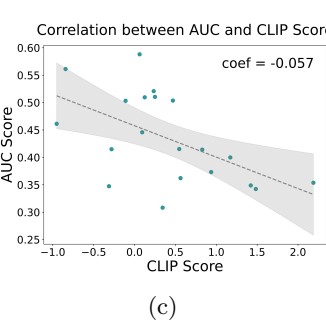

(a)                                        (b)                                        (c)

Figure 10: **CLIP score validation** (whale class). The trends shown in the waterbird class deviate from the ones we observe above. As with the ant class, this can be attributed to potential misalignments in the methodology and the usage of a multiclass classification model. This analysis shows that the validation does not necessarily generalize to the whale class, with the AUC and CLIP scores showing no clear correlation. This suggests that the B2T CLIP score validation generalizes to some classes in the ImageNet dataset.

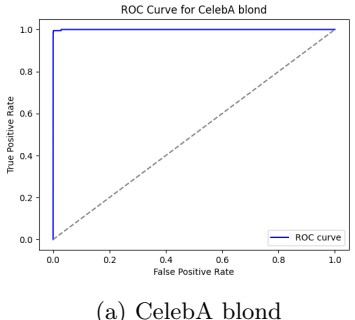

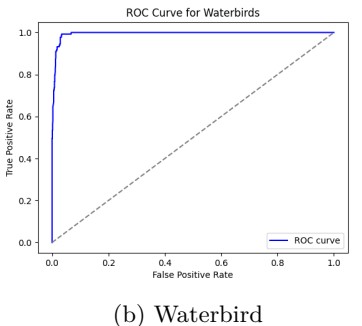

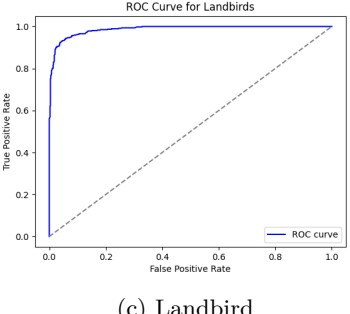

(a) CelebA blond                    (b) Waterbird                    (c) Landbird

Figure 11: ROC curves for (a) CelebA blond (blond male), (b) Waterbird (waterbird on a land background) and (c) Landbird (landbird on water background), illustrating B2T's performance in predicting the correct bias labels for datasets with ground-truth annotations.

## C   Bias elimination & Robustness tests

Table 6: Keywords of the ImageNet ant class according to the ViT-B/16 and ResNet-50 model.

| Bias keyword | ViT-B/16 | ResNet-50 | Match |
|---|---|---|---|
| yellow flower | 2.12 | 2.98 | ✓ |
| green leaf | - | 2.59 | - |
| flower | 1.47 | 1.95 | ✓ |
| glass bowl | - | 1.62 | - |
| glass | - | 1.62 | - |
| red flower | 1.06 | 1.58 | ✓ |
| yellow | 0.84 | 1.45 | ✓ |
| leaf | 0.48 | 1.42 | ✓ |
| flowers | 1.34 | - | - |
| bowl | - | 1.12 | - |
| tree | 0.97 | - | - |
| rock | 0.84 | - | - |
| floor | 0.83 | - | - |
| toilet seat | - | 0.70 | - |
| seat | - | 0.70 | - |
| toilet | - | 0.44 | - |
| close | 0.14 | 0.44 | ✓ |
| bee | - | 0.42 | - |
| garden | 0.39 | - | - |
| world | 0.38 | - | - |
| found | 0.31 | - | - |
| red | - | 0.19 | - |
| bees | - | 0.08 | - |

Table 7: Comparison of the bias keywords discovered by ResNet-50 (i.e., ERM) and the keywords discovered by our trained DRO-B2T for the landbird class. As expected, the DRO model is susceptible to significantly fewer biases.

| Bias keyword | ERM | DRO-B2T | Match |
|---|---|---|---|
| seagull flies | 2.84 | - | - |
| beach | 2.58 | 0.23 | ✓ |
| seagull | 2.47 | 0.22 | ✓ |
| water | 1.25 | - | - |
| lake | 1.03 | - | - |
| rocks | 1.02 | - | - |
| sky | 0.47 | - | - |
| crow | - | 0.36 | - |
| pond | - | 0.33 | - |
| city | 0.28 | - | - |
| dog | 0.22 | - | - |
| park | - | 0.16 | - |

