# OpenReview forum: "On the reproducibility of "Discovering and Mitigating Visual Biases through Keyword Explanation""
_TMLR — Rejected by TMLR_

### Review · Reviewer_m2Qy · 2025-04-01

**Summary Of Contributions:**

The authors present an independent effort to reproduce the findings of Kim et al., which is a method that discovers and labels biases in vision object detection models.  Bias-to-text mitigates two issues in bias detection.  First, it does not rely on multi-labeling for detecting known biases, but can operate without any prior information about biases.  Second, it provides human interpretable bias identification by leveraging existing vision-text language models, to generate bias descriptions for images.

The authors of this paper reaffirm the findings of Kim et al, and extend the original contributions of Kim et al by employing different underlying classification models.  Specifically, the authors provide more concrete details of how to reproduce several of the original experiments of Kim et al, details that were missing or incomplete in the original work.  They discovered additional biases in the ImageNet dataset, and verify that the B2T descriptions of these biases align with prior observations.  Finally, they evaluate the claims of B2T under sampling in the  training data, and under a different base classifier.

**Audience:**

Yes

**Claims And Evidence:**

Yes

**Requested Changes:**

There is a minor inconsistency between the end of section 1 and section 2.  One claim of the existing paper under **Extend the results** is that they:
> test B2T’s performance when the underlying classification model is swapped, ensuring that its effectiveness is not model-dependent.

Yet in section 2, they restate the third claim of the original B2T paper as
> B2T can be deployed with various classifier models, such as Transformer or CNN-based architectures, while still retrieving meaningful bias keywords.

Which calls into question whether this is an extension of the prior work, or independent verification of the original claim. I feel the authors should resolve this by clarifying this point in section 1.  I note that the authors clarify this point somewhat in section 3.2, but it would benefit readers to state it earlier in the paper.

---

When discussing Figure 3 versus Figure 3 of the original B2T paper, it would be more helpful for the authors to comment on just how their findings differ from Figure 3 panel (a) of the original.  I realize that the keywords will be different but it would be helpful for the authors to indicate (on the figure or in the caption) where the other keywords that appear in the original (`forest`, `woods`, `species` `bird`)score in their reproductions.

---

Please add some more motivation in the results section 3.4.3 describing how the authors imagined that B2T might be aversely affected by a low-data regime (see Weaknesses).

---

Please add some citations connecting the discussion at the end of section 5.2, where the authors claim that B2T would be a competitive model for bias discovery, but do not cite any relevant links to the existing body of research in fairness.

**Strengths And Weaknesses:**

### Strengths

This is a reproduction paper, and as such then value that it provides must be measured by whether the authors have made the original work (B2T) more accessible to other researchers, whether they have provided more evidence for the original claims, and whether they have made clear the parts of the original contribution that can be independently substantiated.

In section 2, the authors very clearly restate the original claims of the B2T paper, namely that:
1. B2T can (re)discover both known and new biases of different origin, and explain them using keywords to form a topic in natural language
2. The generated bias keywords from B2T can be used to train a de-biased model using a variant of DRO, and that this usage of DRO surpasses simply using DRO with some ground truth bias labels on relevant metrics
3. B2T works well across different base model architectures

Overall, it is clear that the authors wanted to not only reaffirm the claims of B2T, but also they wanted to clarify the experimental conditions that were underspecified in the original work.

### Weaknesses

The authors could have taken the verification further, by examining the dependency on each element in the original pipeline (Figure 1 of B2T in Kim et al.).  While it is clear that the predicted labels of the base classifier determine its bias, the B2T pipeline relies also on a joint image and text model for captioning (CLIP-Cap), and on a keyword extraction model (YAKE) to work, and the suitability or dependency on both of the latter components is not tested.  This is addressed in the original B2T paper, but deserves a mention in this work on why the authors did not see fit to examine alternatives to the captioning model or keyword extractor.


I am a bit unsure of the motivation for this low-data regime experiment.  In what way do the authors imagine that B2T might fail for small datasets?  I can imagine a few reasons:
1. The base classifier is less accurate on these classes
2. The group bias labels are noisier, meaning that DRO-B2T retraining is less effective

But it's not clear which of these (if any) was intended.  A bit more description of the hypothesized reason for testing this condition would help the paper I think.


I think the paper should spend some more time digging into why DRO-B2T and DRO-GT differ.  Contrasting Table 1 of B2T, with Table 2 of this work shows subtly different results.  In the original paper, DRO with the ground truth is statistically indistinguishable from DRO-B2T.  Yet in this reproduction, there is a clear separation from DRO-B2T from DRO-GT due to the underperformance of DRO-GT here relative to the original paper.  That's an odd discrepancy, and I wouldn't really say that I read this and immediately affirm the claim that DRO-B2T is clearly superior.

Perhaps I missed an explanation in one of the appendices, but I'm really curious as to why this is.  I suspect it is due to the pseudo-labeling strategy of B2T for group labels provides more weak supervision signal than the ground truth labels, which are probably much less abundant.

---

### Review · Reviewer_pWA7 · 2025-04-04

**Summary Of Contributions:**

This paper reproduced the paper "Discovering and Mitigating Visual Biases through Keyword Explanation" paper. Specifically, this paper verified three claims made in the original paper and also provided an extension study on the generality of the B2T model. The paper also includes experiment details complementary to the original paper to facilitate future works.

**Audience:**

Yes

**Claims And Evidence:**

Yes

**Requested Changes:**

Following are some questions and suggestions.

1. What does "The image has a high CLIP similarity score with the image" mean in Fig. 1's caption? What kind of images do the two "images" in the capture refer to, respectively?

2. In Tab. 1, for CelebA and Waterbirds datasets, it looks like the actual and predicted labels are just reversed. Does this phenomenon lead to any insight into the model behavior?

3. Authors can consider including a table in the main paper listing and comparing the claims listed in the original works, the claim verifications by this paper, and the expansion discussions. This table can give the reader a better impression of the paper's contribution and conclusion.

4. Tab. 1 should be closer to the respective discussions on Page 6. Same for Figure 5 (discussion is on page 7).

5. In Tab. 7, authors should consider using different terms to refer to the reproduced and reported results instead of "ours" and "theirs." For example, just saying "reproduced" and "reported in xxx" to avoid any confusion.

6. Better caption on Tab. 3 and 4. For example, what does the "dash" mean, and how should we interpret the numbers? For example, should the number be sparse (e.g., more "dashes" are in the column), or should the number be the lower, the better, or should the numbers between columns be similar? Some discussions are included in 4.2, but the table will be more readable if the respective discussions are also briefly mentioned in the caption.

7. It seems like Sec. 5 repeats many "confirm" and "verification" discussions that overlap with Sec 4. I think the discussion section should focus more on the deeper insights beyond "we confirm results" and should serve as a complementary to the original paper.

8. Typos, like "Figure Figure 3" on page 7.

**Strengths And Weaknesses:**

The reproducibility is quite comprehensive, covering most of the claims made in the paper, except the "different scoring method" part. The comprehensive experiment details and the reconstructed code can provide help to future studies. It is also good to see some expansion discussions. However, those discussions are a bit superficial and do not provide much deeper insights or inspiration than the original work.

---

### Review · Reviewer_CAgw · 2025-04-09

**Summary Of Contributions:**

This report aims to reproduce and extend the results of the Bias-to-Text (B2T) framework (Kim et al., 2024). As a reproducibility report, it includes:
- reproducing the experimental results in support of the original paper’s claims using the released codes as well as own implementation for the missing parts;
- conducting a new biases verification experiment within the ImageNet dataset;
- examining the robustness of the B2T framework to the classification models and the size of datasets;
- providing a more user-friendly codebase.

**Audience:**

Yes

**Claims And Evidence:**

Yes

**Requested Changes:**

1. Related to Weaknesses 2, it would be better to compare the results between the two assignment methods.
2. Related to Weaknesses 3, it would be interesting to conduct a more complete robustness analysis on various ImageNet classes to support the authors’ findings.
3. Please provide some understanding for newly obtained empirical results. For example, it would be better to discuss the reason why the DRO-B2T can outperform the original DRO training with ground-truth bias labels.

**Strengths And Weaknesses:**

Strengths:
1. The paper carefully reproduces the results of a recent paper and shows that the three original claims (bias discovery, bias elimination, and generality) are partially reproducible; the first two claims are reproducible, but the last one shows a somewhat mixed result.
2. The extension of the experiments to examine the robustness of the B2T framework to types of classifiers and size of datasets hints an interesting direction for future work.
3. The paper points out the experimental details that have been missing in the original paper and clearly explains how the authors approached to fill the gap.

Weaknesses:
1. The new benchmark for the biases verification experiment only marginally adds value. The original paper already includes binary classification settings; the new multi-class dataset is transformed into a binary one via a one-versus-rest strategy.
2. In Subsection 3.4.1, paragraph 4, the authors mention two possible assignment strategies, but they simply adopt the mean thresholding without empirical evidence.
3. The robustness analysis focuses only on two ImageNet classes (``ant’’ and ``monastery’’), raising questions about generalizability.
4. In Subsection 3.4.3, the authors ``examine the robustness of the B2T framework by testing the consistency of the discovered keywords’’ without explaining how the consistency is measured. Does the CLIP score of keywords affect the consistency calculation?

---

### Decision · Action_Editor_BcfX · 2025-08-13

**Recommendation:** Reject

**Additional Comments:**

As mentioned in my previous comment, this paper needs to (1) extend the scope of tests for robustness, and (2) conduct a deeper analysis of the DRO-B2T compared to the vanilla DRO to make the claim more convincing. Also, this paper needs minor updates, as the reviewers pointed out.

Although all of these could be addressed by non-significant revision, because there was no rebuttal and no revision, I cannot confirm that this paper is acceptable with a minor revision. Overall, I recommend a major revision.

**Audience:**

Yes

**Audience Explanation:**

Visual biases and spurious correlations are a widely studied topic in machine learning and computer vision. A study working on reproducibility would be helpful for some audience interested in working on visual biases.

**Claims And Evidence:**

No

**Claims Explanation:**

Although all the reviewers provided meaningful and helpful comments, there was no discussion between the reviewers and the authors. I think that the comments from the reviewers could be addressed during the original four-week discussion period. Specifically, as pointed out by the Reviewer CAgw, I think that the current study is too narrow (only considering ant and monastery), which cannot sufficiently support the claim accurately and convincingly.

Also, as pointed out by Reviewer m2Qy, (although the reviewer mentioned that this is a minor concern), the claim made by the current version of the paper is somewhat inconsistent.

Overall, this paper needs to (1) extend the scope of tests for robustness, and (2) conduct a deeper analysis of the DRO-B2T compared to the vanilla DRO to make the claim more convincing. Also, this paper needs minor updates, as the reviewers pointed out.

**Resubmission Of Major Revision:**

The authors may consider submitting a major revision at a later time.